# Hill-Climbing a Real-World VLA in Simulation: Frozen-Backbone Residual Reinforcement Learning for MOLMOACT2

Atharva Kshirsagar     Tirth Gada     Vrushtee Gaikwad

k7agar@gmail.com     gadatirth45@gmail.com     vrushteegaikwad@gmail.com

*Abstract*—Vision–language–action models (VLAs) such as MOLMOACT2 [1] are trained by large-scale imitation learning and deployed on real hardware. As their own authors note, fine-tuned success rates on realistic tasks remain well below the threshold required for dependable use. Reinforcement learning is the natural remedy, but RL on real robots is slow and unsafe, and back-propagating RL through a billion-parameter backbone is unstable and memory-hungry. We study the alternative of hill-climbing a real-world-trained VLA with RL *in simulation*, treating the model as a frozen black box. We freeze MOLMOACT2 and learn a small residual policy that adds a bounded correction to its decoded continuous action chunk; only the residual actor and a critic are trained, online, in ManiSkill [2], and the whole loop fits on one GPU. Our lead experiment is a 14-DoF bimanual YAM packing objects into a box, where a simulated actuator miscalibration ($0.02$ rad joint offset) collapses the frozen policy to $13\%$ success; residual RL recovers it to $33\%$ online while the backbone never receives a gradient. The recovery is seed-dependent (13–33% over three TD3+BC seeds, mean $22\%$, no seed below the base) and holds for a second learner: SAC+BC with the same no-op initialization and base anchor reaches $33\%$ over $45$ held-out episodes. The same recipe, unchanged, replicates on a second embodiment, a single-arm Franka DROID, recovering $10\% \rightarrow 28\%$. Ablations make three design choices concrete: the residual acts on the *decoded action* (decoder-agnostic by construction, though we test one decoder family), it must be *anchored* to the base (naive residual RL drifts to $0\%$, below the base, while no anchored run ever does), and its trust region must be *bounded* (a tighter bound is better for the higher-DoF arm). A cube-stacking ablation marks the boundary: the simulator supports the skill (a privileged base stacks at $25\%$) but the real-trained VLA does not transfer into sim ($0\%$), so sim RL hill-climbs skills the policy already partially performs but cannot cross an unmatched domain gap. All results are in simulation against simulator-injected perturbations; validating that the learned corrections transfer to physical hardware is future work. We release the code as a drop-in extension of the public MOLMOACT2 evaluation stack.

## I. INTRODUCTION

Vision–language–action models (VLAs) such as MOLMOACT2 [1], $\pi_0$ [3], and OpenVLA [4] are trained by large-scale *imitation learning* and ship as strong real-world manipulation policies. MOLMOACT2, for example, couples a flow-matching continuous-action expert to the Molmo2-ER vision–language backbone [1] and reports $87.1\%$ mean success on the real Franka/DROID suite. Yet imitation alone is rarely

enough: deployment surfaces a long tail of systematic errors, slightly mis-timed grasps, consistent spatial offsets, premature gripper releases. The authors say as much: even after task-specific fine-tuning, success rates on realistic tasks remain below the threshold required for dependable use [1].

Reinforcement learning is the natural remedy, but RL on real robots is slow, reset-hungry, and unsafe. Applying RL to the foundation model itself is also awkward: the backbones are billions of parameters, their action heads span autoregressive, diffusion, and flow-matching decoders (so each model needs a bespoke RL formulation), and back-propagating a policy gradient through a frozen-then-unfrozen VLA is unstable and memory-intensive. We therefore study a different setting: hill-climbing a real-world VLA with reinforcement learning entirely *in simulation*, where rollouts are cheap, resets are free, and privileged state gives a dense reward. This is a sim-to-real question in reverse, since a model trained on real data is improved in simulation, putting the real/sim gap front and center.

Within this setting we keep the recipe minimal and leave the backbone frozen. We treat the foundation model as a fixed black box that, given an observation and a language instruction, emits a chunk of decoded continuous actions. We then learn a small residual policy that outputs a bounded correction $\Delta a$ and execute $a = a_{\text{base}} + \Delta a$ (Fig. 1). Only the residual actor and a value critic are trained, with an off-the-shelf off-policy algorithm (TD3 [5]). The foundation model is queried purely in inference mode, behind an HTTP server on the same GPU, so a single A100 suffices and no part of the recipe depends on the backbone's internals.

The key design choice is *where* the residual acts. Acting on the decoded action, rather than on backbone logits or flow noise, makes the method backbone-agnostic *by construction*: it only ever sees a vector of continuous actions, which autoregressive/FAST [6], diffusion, and flow-matching decoders all ultimately emit, and any standard actor–critic method can train it. We are explicit about what is demonstrated versus argued: our experiments use one backbone family (MOLMOACT2's flow-matching decoder, on two embodiment checkpoints) and two learners (TD3+BC and SAC+BC, which recover the same task equally well, Sec. V-A), so algorithm-generality has

direct support while decoder-generality remains a structural argument, not an empirical result of this paper. This is the same inductive bias that makes classical residual RL for robot control [7] sample-efficient: learn a small correction on top of a strong base rather than a policy from scratch, now applied to a modern VLA.

Rather than spreading thin across tasks, we build the argument on one *lead* deployment experiment, designed so that each ablation isolates one of the method's design choices, and replicate the recipe unchanged on a second embodiment (single-arm Franka DROID, Sec. V-B).

**Contributions.**

- A minimal recipe for improving a real-world VLA with RL in simulation, freezing the backbone entirely: a bounded residual on the decoded action chunk plus an off-the-shelf critic, released as a drop-in extension of the public MOLMOACT2 `sim_eval`/ManiSkill stack.
- A single deployment experiment, bimanual-YAM packing under actuator miscalibration, that recovers the frozen policy from 13% to 33% online and, through its ablations, pins down the three choices that matter: residual-on-decoded-action, base-anchoring (TD3+BC; naive RL drops *below* the base), and a bounded trust region (tighter for higher DoF).
- A cube-stacking ablation that maps the method's boundary. By splitting a single "0% in sim" into two questions, whether the simulator *supports* the skill (a privileged base stacks at 25%) and whether the real-trained VLA *transfers* into sim (0%), it shows the binding constraint is the real→sim gap, not sim RL.

## II. RELATED WORK

**Online RL for VLAs.** A growing line fine-tunes VLAs with online RL: SimpleVLA-RL [8], iRe-VLA [9], $\pi$RL [10] (online RL for flow-based VLAs), and $\pi_{0.6}^*$ [11] all *update backbone weights* (full or LoRA) and are largely decoder-specific. Xiao et al. [12] also use residual RL, but to *generate* self-improvement data that is folded back into the policy weights. Our recipe instead keeps the backbone strictly frozen and inference-only. The closest neighbours share the "don't touch the whole backbone" spirit but differ in *where* they intervene: **PA-RL** [13] optimizes actions and distills them into the base policy (we never distill or modify the backbone); **EXPO** [14] introduces a residual edit-actor with a value critic (we generalize the residual idea to a frozen *foundation* model in simulation, and show it is decoder-agnostic); and **DSRL** [15] steers diffusion policies in their *latent noise* space, which is decoder-specific, whereas our residual acts on the decoded action and should in principle extend to discrete/FAST [6] models too. Residual RL on a hand-designed base controller is classical [7]; we apply it to a learned foundation model. Sample-efficient real-world RL (HIL-SERL [16]) is complementary and could replace our learner. **Sim-to-real for VLAs.** MOLMOACT2 [1] itself is a large-scale *imitation* model trained on real and teleoperated data (including a 720-hour bimanual YAM corpus) and reports

no RL or simulation fine-tuning; we treat it as a fixed real-world policy and study what RL *in simulation* adds on top. We evaluate in ManiSkill [2] (SAPIEN), following the real2sim evaluation protocol that SIMPLER [17] established for VLAs; MOLMOACT2 was never tuned in simulation, so its zero-shot sim behavior already reflects a real→sim shift.

## III. METHOD

### A. Setup and notation

A frozen foundation model $\pi_{\text{base}}$ maps an observation $o_t$ (RGB cameras + proprioceptive state) and a fixed language instruction $\ell$ to a chunk of $H$ decoded continuous actions $\mathbf{a}_{t:t+H}^{\text{base}} = \pi_{\text{base}}(o_t, \ell)$. Each action is $d=14$-dimensional for the bimanual YAM (two arms of 6 joints + a gripper each), produced by a flow-matching action expert; the chunk is executed open-loop and re-queried when exhausted, exactly as in the public evaluator. Crucially, our method only ever observes the *output* vector $\mathbf{a}^{\text{base}}$, not the model's internals.

### B. Residual policy on the decoded action

We learn a deterministic residual actor $\mu_\theta(s_t) \in [-1,1]^d$ and execute

$$a_t = a_t^{\text{base}} + \alpha \cdot \mu_\theta(s_t), \qquad (1)$$

where $\alpha$ is a fixed per-dimension scale bounding the correction (the *residual scale*) and the actor state is the compact vector

$$s_t = \left[ \text{proprio}(o_t) ; a_t^{\text{base}} \right] \in \mathbb{R}^{2d}. \qquad (2)$$

The residual sees *no pixels*: all perceptual information reaches it through $a_t^{\text{base}}$, the backbone's own decoded intent. This is what keeps the residual small and backbone-agnostic: it corrects the base controller's behavior rather than re-solving perception. A bounded $\alpha$ guarantees the executed action stays in a trust region around the foundation model, so at $\alpha\to 0$ behavior reduces exactly to the zero-shot base.

### C. Learner: TD3, anchored to the base

We train $\mu_\theta$ and a twin-critic $Q_\phi$ with TD3 [5] (clipped double-Q, delayed and smoothed target policy, Gaussian exploration). The RL transition is $(s_t, \Delta a_t, r_t, s_{t+1})$ where the "action" is the residual in the normalized $[-1,1]$ space of Eq. (1); the foundation model contributes $a^{\text{base}}$ to both $s_t$ and the executed command but receives no gradient. We bootstrap only on genuine task termination, treating the time limit as truncation.

**Naive residual RL collapses, and why.** A deterministic actor maximizing $Q$ is free to push the residual to the boundary of its $\alpha$-box. Early in training the critic is only accurate near $\Delta a \approx 0$ (where data lives), and overestimates $Q$ in the unexplored corners; the actor chases that error and the executed policy drifts *below* the zero-shot base, collapsing to 0%. Two ingredients fix this and are essential, not cosmetic: (i) a **no-op initialization**, we zero the actor's last layer so the initial residual is exactly 0 and rollouts begin at the base; and (ii) a **base anchor**, a behavior-cloning term that regularizes the residual toward 0 (TD3+BC [18] with the "behavior" being

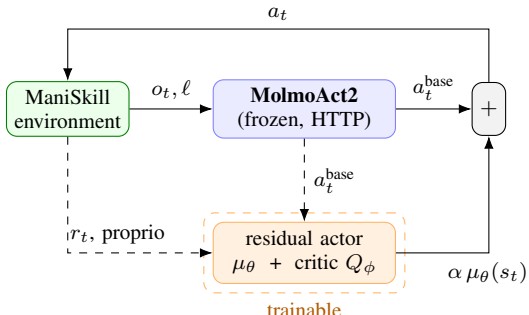

Fig. 1. **Frozen-backbone residual RL.** MolmoAct2 (blue, frozen, served over HTTP) decodes a continuous action chunk $a_t^{\text{base}}$. A small residual actor (orange) adds a bounded correction $\alpha\,\mu_\theta(s_t)$ in the *decoded action* space; the sum is executed. Only the residual actor and critic are trained (TD3+BC or SAC+BC). No gradient ever enters the backbone.

the frozen base), so the actor leaves the base only where the advantage clearly outweighs the anchor:

$$\mathcal{L}_{\text{actor}} = -\,\lambda\,\mathbb{E}\big[Q_\phi(s, \mu_\theta(s))\big]\ +\ \mathbb{E}\big[\|\mu_\theta(s)\|^2\big], \quad \lambda = \tfrac{\beta}{|Q|}. \tag{3}$$

With both, training starts at the base and does not fall below it. Any other continuous-control learner drops in unchanged, keeping the same two ingredients; we verify this with SAC [19]+BC in Sec. V-A.

### D. Dense reward that preserves the success metric

The shipped benchmark reward is essentially sparse (objects-in-box count). For a learnable critic we add a staged shaping reward (reach→grasp →transport→place) computed from privileged simulator state, while keeping the *exact* success definition unchanged, so reported success rates remain directly comparable to the zero-shot baseline. Shaping is used only for the RL reward channel; it does not alter dynamics, observations, or the success test.

## IV. EXPERIMENTAL SETUP

We build the case on one *lead* experiment, designed so that each ablation isolates one of the method's choices, and replicate the recipe unchanged on a second embodiment (Sec. V-B). All experiments, including the deployment error itself, are in simulation (Sec. VI).

**The frozen VLA.** MOLMOACT2 [1] connects a flow-matching continuous-action expert (depth-matched to the VLM at 36 layers, via per-layer key–value conditioning) to the Molmo2-ER vision–language backbone (with a SigLIP2 vision encoder), trained by imitation on real and teleoperated data including a 720-hour, 34.5k-demo bimanual YAM corpus spanning 28 real tasks. We use its public MOLMOACT2-BimanualYAM checkpoint unmodified.

**Embodiment, task, and benchmark.** The embodiment is the *bimanual* YAM, a 14-DoF dual-arm robot (two 6-DoF arms + two grippers) driven from three RGB views (top/left/right). The task is PutEverythingInBox: place a Lego duplo and a tennis ball into a box (success = both objects in the box), a simulation analog of MOLMOACT2's real "block in box" task

TABLE I
THE LEAD EXPERIMENT: BIMANUAL YAM UNDER 0.02 RAD ACTUATOR MISCALIBRATION (HELD-OUT, $n{=}15$, 1500-STEP HORIZON; TRAINING SEED 0 — SEE TABLE II FOR SEED AND EVALUATION VARIANCE). MOLMOACT2-BIMANUALYAM FROZEN IN ALL ROWS. EACH ROW ISOLATES ONE DESIGN CHOICE: ANCHORING (VS. NAIVE) AND TRUST-REGION SIZE ($\alpha$).

| Method (14-DoF dual-arm) | Success (%) |
|---|---|
| MolmoAct2-BimanualYAM (zero-shot, miscalibrated) | 13 |
| + residual RL, naive TD3 (no anchor) | 0 |
| + residual RL, $\alpha{=}0.10$ | 27 |
| + residual RL, $\alpha{=}0.05$ (ours) | **33** |

(33.3% real success on a low-cost arm). The model is served in `bfloat16` ($<$16 GB) behind the public FastAPI `/act` server, and backbone, residual, and simulator share a single A100 with headless Vulkan rendering. One benchmarking caveat shapes the bimanual setup. At ManiSkill's default 400-step horizon the dual-arm task scores 0% zero-shot, but only because the slower arms cannot place both objects within the allotted steps, a *horizon artifact* rather than a capability gap. We therefore evaluate at a 1500-step horizon, at which the base recovers.

**The deployment shift: actuator miscalibration.** To model a frozen policy deployed on a robot whose joint zeros differ from its training embodiment, we add a fixed bias to every commanded arm-joint target inside the controller. This is a *systematic, state-independent* error, the prototypical case an action-space residual should be able to cancel: a 0.02 rad offset drops zero-shot success to 13%. Dynamics, cameras, objects, and the success test are identical to the nominal task; only the bias changes.

**Protocol.** The residual is trained online for 12,000 environment steps on training seeds, then evaluated on 15 *held-out* seeds (disjoint from training) with the deterministic residual (no exploration noise) against the zero-shot base under identical seeds and chunking. We use TD3+BC with 8 gradient updates per environment step (gradient steps are cheap relative to a backbone query, so a high update-to-data ratio maximizes sample-efficiency), $\gamma{=}0.99$ (Tab. V).

## V. RESULTS

**The residual recovers the miscalibrated bimanual policy.** Table I reports held-out success on the 14-DoF bimanual YAM under the 0.02 rad actuator miscalibration ($n{=}15$, 1500-step horizon), with MOLMOACT2-BimanualYAM frozen in every row. The frozen base scores 13%; the residual, trained online while the backbone never receives a gradient, recovers it to 33% at $\alpha{=}0.05$, learning a near-constant counter-offset that cancels the systematic bias. The same table is, by construction, three ablations at once, one per design choice.

**Choice 1: anchor to the base.** Removing the anchor is catastrophic. Naive TD3, with no behavior-cloning term and a randomly initialized actor, collapses to 0%, *below* the already-degraded base (Table I, row 2): the un-anchored residual exploits an overestimated critic in unexplored corners of the $\alpha$-box and drifts away from the base, while the base-anchored

residual (TD3+BC, with no-op initialization) starts at the base and recovers task success, reaching 33% (Table I, row 4). This is the single most load-bearing ingredient in the recipe.

**Choice 2: bound the trust region.** The residual scale $\alpha$ sets how far the executed action may stray from the frozen base. On the 14-DoF arm, the *tighter* bound wins: $\alpha=0.05$ reaches 33% while the looser $\alpha=0.10$ reaches only 27% (Table I). With more action dimensions there is more room to over-perturb, so the trust region must shrink, the same recipe on the 8-D single-arm Franka DROID prefers the looser $\alpha=0.10$ (Sec. V-B), confirming the cross-embodiment trend.

**Choice 3 (scope): act in decoded-action space, on a base with headroom.** The recovery works precisely because mis-calibration is a systematic, state-independent error expressible in decoded-action space, on a base that already partially solves the task. Where neither holds, the residual does not help: it is pixel-free, so it cannot relocalize objects the backbone mislocalizes under perceptual shift, and on a *strong* nominal base (DROID, 45%, $n=20$) it *degrades* success to 20% at $\alpha=0.10$, with no systematic error to cancel, its corrections only perturb an already-good policy. The stacking ablation (Sec. V-C) makes the complementary boundary, *skill coverage*, concrete.

### A. Seeds, evaluation noise, and a second learner

We calibrate how large and how repeatable the recovery is (Table II).

**Seed variance.** We trained two more TD3+BC residuals (seeds 1 and 2) with the identical protocol and evaluated each on 30 held-out episodes (two 15-episode passes over disjoint held-out seed sets). With the seed-0 run of Table I, the three seeds reach 33/20/13% against the 13% base, mean 22%. The recovery is real but seed-dependent: two of three seeds improve substantially, the third only matches the base. Crucially, no anchored seed falls *below* the base while naive TD3 collapses to 0%: even when RL fails to find an improvement, the anchor bounds the downside at the base policy.

**Evaluation noise.** Re-evaluating a single fixed SAC checkpoint on the *same* 15 held-out episodes yields 47% and then 20%; a third pass on 15 fresh held-out episodes yields 33%. The residual is deterministic at evaluation, but the backbone's flow-matching decoder samples noise at inference, so episode outcomes are stochastic, and at $n=15$ the binomial standard error alone is $\approx 12$ points. The numbers in this subsection therefore pool 30–45 episodes per run; single-pass $n=15$ numbers elsewhere in the paper (including the seed-0 row of Table I) carry that uncertainty.

**A second learner: SAC+BC.** Swapping TD3 for SAC while keeping the two load-bearing ingredients, the no-op initialization and the Q-normalized base anchor, and the identical training protocol, recovers 33% (15/45 episodes), matching the best TD3 seed with the tightest-estimated number in this paper. The recovery is not specific to one RL algorithm.

TABLE II
SEED AND EVALUATION VARIANCE, AND A SECOND LEARNER
(BIMANUAL YAM UNDER 0.02 RAD MISCALIBRATION; DETERMINISTIC RESIDUAL ON HELD-OUT EPISODES; BACKBONE FROZEN).

| Method | Success (%) | $n$ |
|---|---|---|
| MolmoAct2-BimanualYAM (zero-shot) | 13 | 15 |
| + naive TD3 (no anchor) | 0 | 15 |
| + TD3+BC, seed 0 | 33 | 15 |
| + TD3+BC, seed 1 | 20 | 30 |
| + TD3+BC, seed 2 | 13 | 30 |
| + TD3+BC, mean over seeds | 22 | – |
| + SAC+BC | **33** | 45 |

**MolmoAct2-DROID rollout: put-everything-in-box**

Fig. 2. **The second embodiment: single-arm Franka DROID in ManiSkill.** *Top:* rollout of the frozen MOLMOACT2-DROID policy on `put-everything-in-box`. *Bottom:* the external and wrist RGB views the backbone consumes; the residual never sees them, it observes only proprioception and the decoded action chunk.

### B. Cross-embodiment replication: single-arm Franka DROID

The same recipe, unchanged, holds on a second robot and a decoder-identical backbone: MOLMOACT2-DROID (Franka FR3 + Robotiq, 8-D action, external+wrist views, Fig. 3) on the same packing task, under a 0.008 rad actuator miscalibration. Table III repeats the YAM story: the bias drops zero-shot success to 10%; naive TD3 again collapses to 0%; the base-anchored residual recovers to 28% (mean of 2 training seeds; both beat the base, at 40% and 15%, which also gives a first look at seed variance). The best trust region is the *looser* $\alpha=0.10$, consistent with the lower action dimensionality (Choice 2): across a single-seed sweep, $\alpha=0.02/0.05/0.10$ reach 25/10/40%. We caution that the non-monotonicity of that sweep is within single-seed noise at $n=20$; the robust pattern, seen on both embodiments, is that the best $\alpha$ shrinks as DoF grows.

### C. Ablation: the sim-to-real gap, not sim-RL, is the wall

The miscalibration experiment hill-climbs a skill the VLA *already* partially performs. A natural question is whether sim RL can instead originate a skill the VLA lacks. We answer it on cube-stacking, MOLMOACT2's *weakest* reported real skill ("stack blocks," 20% on a low-cost arm), and in doing so separate two questions a single "0% in sim" would conflate: (Q1) *does the simulator support the skill?* and (Q2) *does the real-trained VLA transfer into sim?* We use `BimanualYAMStackCube` (stack a red cube on a green cube; success = stacked, released, static), 12 held-out seeds at the 1500-step horizon for the VLA conditions (Table IV).

**Q1: the simulator supports stacking.** A privileged scripted

TABLE III

**Cross-embodiment replication: single-arm Franka DROID** under $0.008$ rad actuator miscalibration (held-out, $n=20$). MolmoAct2-DROID frozen in all rows.

| Method (8-DoF single-arm) | Success (%) |
|---|---|
| MolmoAct2-DROID (zero-shot, miscalibrated) | 10 |
| + residual RL, naive TD3 (no anchor) | 0 |
| + residual RL, $\alpha{=}0.02$ | 25 |
| + residual RL, $\alpha{=}0.05$ | 10 |
| + residual RL, $\alpha{=}0.10$ (mean of 2 seeds) | **28** |

TABLE IV

**Stacking ablation: simulator support (Q1) vs. real→sim transfer (Q2).** Held-out success (%). A privileged base stacks well above zero, so the simulator supports the skill; the frozen MolmoAct2-BimanualYAM never completes a stack, and a residual on a $0\%$ base cannot create one.

| Condition | Skill source | Success (%) |
|---|---|---|
| *Q1: does the simulator support the skill?* | | |
| privileged scripted base | ground-truth poses | **25** |
| *Q2: does the real-trained VLA transfer to sim?* | | |
| generic MolmoAct2-BimanualYAM | — | 0 |
| + fine-tune (44 real), zero-shot sim | real demos | 0 |
| + residual RL (sim) | sim RL | 0 |

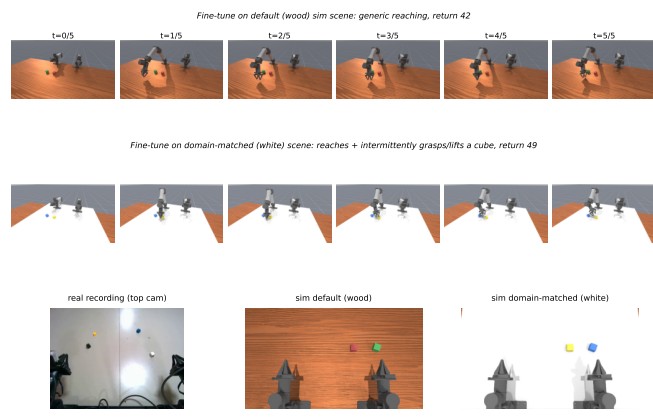

Fine-tune on default (wood) sim scene: generic reaching, return 42

Fine-tune on domain-matched (white) scene: reaches + intermittently grasps/lifts a cube, return 49

real recording (top cam)  sim default (wood)  sim domain-matched (white)

Fig. 3. **Bimanual cube-stacking and the real→sim gap.** Same task and seed. *Top:* the generic base reaches generically and never grasps a cube. *Middle:* the checkpoint fine-tuned on 44 real "stack the cubes" demonstrations drives its gripper onto a cube (visibly stacking-oriented), but the grasp does not close in SAPIEN and no stack completes (0%). *Bottom:* the top/left/right RGB the policy consumes in sim, synthetic textures and lighting far from its real training data.

controller with direct access to cube poses stacks at $25\%$ ($n{=}20$), so the task, contact dynamics, and success test are all achievable in sim. Any VLA $0\%$ below is therefore a *transfer failure*, not a simulator limitation.

**Q2: the real-trained VLA does not transfer.** The generic frozen base scores $0\%$ (stacking is outside its pick-and-place repertoire). A checkpoint fine-tuned from the same backbone on 44 *real* "stack the cubes" demonstrations also scores $0\%$ zero-shot in sim, but its failure is qualitatively different: it drives the gripper directly onto a cube and attempts a grasp, visibly stacking-oriented behavior the generic base never shows, yet the grasp does not close in SAPIEN. The skill is present in the checkpoint but does not survive the real→sim gap; a residual on a $0\%$ base then has nothing to anchor to and also scores $0\%$. A domain-matching ablation (white table, matched lighting/colors; Fig. 4) recovers reaching and *intermittent grasp-and-lift* (to $6.2$ cm) but still completes no stack. The gap is thus two-layered, an *appearance* gap (partially closable by visual matching) and a *placement/contact-dynamics* gap that it does not close.

**Takeaway.** The experiments bracket the method. In simulation, RL hill-climbs a real-world-trained VLA on systematic deployment errors and skills it *already partially performs* (miscalibration: $13\%{\rightarrow}33\%$ on YAM, $10\%{\rightarrow}28\%$ on DROID), but it neither relocalizes perception nor originates, or validates, a skill across an unmatched domain gap (stacking: $0\%$, bounded by real→sim transfer rather than by sim-RL).

## VI. Limitations

Three limitations bound our claims. **No real-robot validation.** Although the backbone is a real-world-trained VLA, every result in this paper is obtained in simulation, and the

deployment error we recover from is itself simulator-injected: a fixed, state-independent joint offset, far simpler than the drifting, state-dependent error distributions of physical hardware (backlash, wear, thermal drift, perceptual error). What the experiments establish is that a frozen VLA degraded by a systematic actuation error can be recovered by a bounded residual trained online in simulation; whether such a residual transfers to a physical robot is untested and is the immediate next step. **Statistical power.** Sec. V-A measures the two noise sources directly: recovery is seed-dependent ($33/20/13\%$ across three TD3+BC seeds, mean $22\%$, none below the base), and a 15-episode evaluation of a fixed checkpoint carries $\approx$12-point noise (repeat passes of one checkpoint: $47\%$ vs. $20\%$), which also bounds how precisely the single-pass numbers elsewhere in the paper should be read. The load-bearing evidence is the ordering, consistent across every run and both embodiments: naive residual RL falls below the base, and the anchored residual never does. **Decoder-generality argued, not demonstrated.** Acting on the decoded action makes the recipe decoder- and algorithm-agnostic *by construction*; two learners (TD3+BC, SAC+BC) now support the algorithm half empirically, but we still validate only one backbone family (MolmoAct2's flow-matching decoder); the residual is also pixel-free, so it cannot correct perceptual failures (Choice 3).

## VII. Conclusion

We asked what reinforcement learning *in simulation* can do for a real-world-trained, imitation-based VLA, treating MolmoAct2 as a frozen black box and learning only a small bounded residual on its *decoded* action chunk with an off-the-shelf critic, a recipe that is decoder-agnostic by construction and fits on a single GPU. One lead deployment experiment carries the argument: on a 14-DoF bimanual YAM, sim RL recovers a simulated systematic actuator miscalibration from $13\%$ to $33\%$ (best of three seeds; mean $22\%$, no seed below

the base), a second learner (SAC+BC) reaches $33\%$ over $45$ episodes, and the ablations pin down the choices that matter, a base anchor (naive RL drifts *below* the base), a bounded trust region (tighter for higher DoF), and a decoded-action intervention on a base with headroom; the recipe replicates unchanged on a single-arm Franka DROID ($10\%\rightarrow28\%$). A cube-stacking ablation marks the boundary: the simulator supports the skill (a privileged base hits $25\%$) but the real-trained VLA does not transfer into sim, so RL has a $0\%$ start with nothing to hill-climb. The takeaway: simulation is a cheap, safe place to hill-climb a real-world-trained policy on systematic deployment errors and skills it already partially performs, but it neither relocalizes perception nor originates, or validates, a skill across an unmatched domain gap, and whether the learned corrections survive contact with physical hardware remains to be shown (Sec. VI). Natural extensions are real-robot validation of a sim-trained residual, a perception-conditioned residual, a second decoder family, and closing the real$\rightarrow$sim dynamics gap so a real-trained skill can be hill-climbed in sim and returned.

## APPENDIX

### A. Implementation and hyperparameters

The foundation model is served by the public MOLMOACT2 FastAPI `/act` server (`bfloat16`, flow-matching action expert) and queried over loopback HTTP; the residual learner runs in the same process group on the same A100. The residual actor and twin critic are 2-layer MLPs (256 hidden units, ReLU); the actor output passes through `tanh` and is scaled by $\alpha$; the actor's last layer is zero-initialized so the initial residual is exactly 0. States are normalized with running mean/variance statistics. We run 8 (YAM) or 4 (DROID) gradient updates per environment step after a short warm-up. Episodes use a 1500-step limit (to avoid the horizon artifact); the action chunk is executed and re-queried exactly as in the zero-shot evaluator, identically for base and residual conditions, so the only difference between conditions is the additive residual.

TABLE V
TD3 / SAC RESIDUAL HYPERPARAMETERS.

| Hyperparameter | Value |
|---|---|
| discount $\gamma$ | 0.99 |
| target smoothing $\tau$ | 0.005 |
| actor / critic learning rate | $3\times10^{-4}$ |
| batch size | 512 |
| replay capacity | $2\times10^{5}$ |
| gradient updates / env step | 8 (YAM) / 4 (DROID) |
| exploration noise (std) | 0.3 (YAM) / 0.2 (DROID) |
| target policy noise / clip (TD3) | 0.2 / 0.5 |
| policy update delay (TD3) | 2 |
| BC anchor weight $\beta$ | 2.5 |
| initial policy std (SAC) | 0.3 |
| entropy temperature (SAC) | auto-tuned, target $-d$ |
| residual scale $\alpha$ | 0.05 (YAM) / 0.10 (DROID) |
| hidden sizes | $256 \times 256$ |
| state dimension | $2d$: 28 (YAM) / 16 (DROID) |
| action (residual) dimension | $d$: 14 (YAM) / 8 (DROID) |
| warm-up steps | 1500 (YAM) / 2000 (DROID) |

### B. Why act on the decoded action?

Three common intervention points exist for steering a frozen VLA: the backbone's token logits, the decoder's internal latent (e.g. diffusion/flow noise, as in DSRL [15]), and the *decoded* continuous action. Only the last is shared by every decoder family: an autoregressive/FAST [6] model, a diffusion policy, and MOLMOACT2's flow expert all ultimately emit a continuous action vector. Operating there costs us access to the decoder's internal degrees of freedom, but buys decoder- and algorithm-independence and a hard trust-region guarantee via the bound $\alpha$. For online fine-tuning of heterogeneous robot foundation models, we argue that trade is worth making.

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
