# OpenReview forum: "Hill-Climbing a Real-World VLA in Simulation: Frozen-Backbone Residual Reinforcement Learning for MOLMOACT2"
_roboticsfoundation.org/RSS/2026/Workshop/RL4VLA — RL4VLA_

### Official Review · Reviewer_KHMa · 2026-06-28
**A paper proposing a minimal recipe for improving a VLA with RL in simulation for the real-world**

**Rating:** 5
**Confidence:** 5

**Review:**

Relevance: Directly on-topic for the workshop: using RL for VLA post-training, with a thoughtful treatment of the sim/real boundary.

Strengths:
The core idea is clean and practical. A bounded residual on the decoded action is decoder-agnostic by construction and keeps the backbone frozen and inference-only, fitting on a single GPU.

The no-op initialization is a well-justified fix for the known boundary-drift failure of a deterministic actor maximizing an overestimated critic.

Weaknesses:
There is no real-robot experiment, despite the framing. The title and abstract have "real-world" and "sim-to-real" in them, but every result is in simulation. The central practical claim, improving a real-world VLA, is therefore demonstrated entirely in simulation against a sim-injected perturbation, and this should be made more explic.

The paper makes the claim that the method is decoder/algorithm-agnostic yet they fix the RL algorithm used (TD3) and the VLA/decoder used (MolmoAct2/flow-matching). The paper should either not explicitly claim that their method works across any decoder/RL algorithm or add these experiments.

---

### Official Review · Reviewer_BgRv · 2026-06-30
**Hill-Climbing a Real-World VLA in Simulation: Sim-to-Real Residual Reinforcement Learning for MolmoAct2**

**Rating:** 6
**Confidence:** 4

**Review:**

This paper proposes a minimal post-training recipe for real-world VLAs, i.e., keep the VLA backbone frozen and learn a small bounded residual on its decoded continuous action chunk using TD3+BC in simulation. The authors validate three key design choices (acting on decoded actions, base anchoring, and bounded trust regions) through a single 14-DoF bimanual YAM experiment under actuator miscalibration.

pros:
- The method is simple and practical.
- Acting on decoded actions makes the approach decoder-agnostic.

cons:
- Limited empirical evidence. The main result is single-seed with only 15 held-out episodes and covers one task and one synthetic perturbation, making the gains indicative rather than conclusive.
- No real-robot validation. The paper claims to address sim-to-real RL, but the residual is trained entirely in simulation and never tested on physical hardware. It shows that simulation can fix a simulation error, not that the fix transfers to reality. Since real deployment errors are typically more complex than the fixed 0.02 rad miscalibration assumed in the paper, it is hard to judge if the method truly scale to real deployment.
- The residual has no visual input, so it cannot correct perception failures.

---

### Decision · Program_Chairs · 2026-07-03

**Decision:**

Accept

**Comment:**

This paper proposes a simple post-training method for VLAs that learns a bounded residual on decoded actions using TD3+BC while keeping the backbone frozen. The reviewers found the idea practical, but their main concern is the limited empirical evidence, as all experiments are conducted in simulation on a single task with one perturbation setting and no real-robot validation. Therefore, the claims about real-world applicability and broad decoder/algorithm generality are not yet fully supported. Nevertheless, the paper is a valuable contribution to the workshop. For the camera-ready version, the authors should clearly acknowledge these limitations, soften the real-world claims, and, if feasible, include a small additional simulation ablation (e.g., more seeds or perturbation settings), while leaving real-robot validation as future work.